# Comprehensive Transcriptomic Analysis of Heterotrophic Nitrifying Bacterium *Klebsiella* sp. TN-10 in Response to Nitrogen Stress

**DOI:** 10.3390/microorganisms10020353

**Published:** 2022-02-03

**Authors:** Dan Li, Mingquan Huang, Shirong Dong, Yao Jin, Rongqing Zhou, Chongde Wu

**Affiliations:** 1College of Biomass Science and Engineering, Sichuan University, Chengdu 610065, China; 2016223080022@stu.scu.edu.cn (D.L.); yaojin12@scu.edu.cn (Y.J.); zhourqing@scu.edu.cn (R.Z.); 2Beijing Laboratory of Food Quality and Safety, Beijing Technology and Business University, Beijing 100048, China; 3Sichuan Fansaoguang Food Group Co., Ltd, Chengdu 611732, China; dsr@fansaoguangfood.com

**Keywords:** heterotrophic nitrification, *Klebsiella* sp. TN-10, transcriptome, regulation networks, nitrogen stress

## Abstract

*Klebsiella* sp. TN-10, a heterotrophic nitrifying bacterium, showed excellent nitrification ability under nitrogen stress. The strain was cultured under different nitrogen stress levels, including ammonium sulfate 0.5, 2.5, and 5 g/L, and samples were titled group-L, group-M, and group-H, respectively. In these three groups, the removed total nitrogen was 70.28, 118.33, and 157.18 mg/L after 12 h of cultivation, respectively. An RNA-Seq transcriptome analysis was used to describe key regulatory networks in response to nitrogen stress. The GO functional enrichment and KEGG enrichment analyses showed that differentially expressed genes (DEGs) participated in more pathways under higher nitrogen stress (group-H). Carbohydrate metabolism and amino acid metabolism were the most abundant subcategories, which meant these pathways were significantly influenced by nitrogen stress and could be related to nitrogen removal. In the nitrogen cycle, up-regulated gene2311 (*narK*, encodes major facilitator superfamily transporter) may accelerate the entry of nitrogen into the cells and subsequently contribute to the nitrogen utilization. In addition, the up-regulation of gene2312 (*narG*), gene2313 (*narH*), and gene2315 (*narH*) may accelerate denitrification pathways and facilitate nitrogen removal. The results presented in this study may play a pivotal role in understanding the regulation networks of the nitrifying bacterium TN-10 under nitrogen stress.

## 1. Introduction

As ammonia waste streams are massively discharged from industrial, agricultural, and organism metabolism, making nitrogen pollution has become a considerable environmental problem [1,2]. Additionally, it is difficult to treat the high-strength ammonia nitrogen wastewater that comes from landfills, dyeing factories, tanneries, and chemical plants [3,4]. Currently, several novel heterotrophic nitrifying bacteria with ammonia tolerance abilities have been isolated from natural environments, such as *Acinetobacter junii* YB [5], *Bacillus subtilis* A1 [6], and *Providencia rettgeri* YL [7]. These heterotrophic nitrifying bacteria have distinct advantages, such as utilizing organic substances, growing rapidly, and tolerating oxygen, which indicate a promising application in nitrogen wastewater treatment [8,9]. Many researchers have reported the basic information of these nitrifiers, including possible nitrifying-denitrifying pathways, key enzymes, crucial genes, whole genomes, etc. [10,11]. However, little remains known about the physiology of nitrifying bacteria based on traditional measurement methods.

Next-generation sequencing technologies use deep-sequencing technologies to describe the whole genome or transcriptomes [12]. RNA sequencing (RNA-Seq) technology has been successfully applied to unravel RNA expressions and study the differentially expressed genes (DEGs) under different treatments. At present, a small number of studies have utilized RNA-Seq technology to explore how nitrifiers or denitrifiers modify their physiology during nitrogen starvation or nitrogen stress. For example, it was reported that the growth of *Nitrobacter winogradskyi* Nb-255, a nitrite-oxidizing bacterium, was depressed under NH_4_^+^ concentrations higher than 35 mM, but cells grew well under NH_4_^+^ concentrations below 25 mM [13]. *N. winogradskyi* significantly adjusted gene expression at higher NH_4_^+^ concentrations, such as the enhancement in glutamate synthase or depression of NO_2_^−^ assimilatory, glycogen, or biofilm and motility. Additionally, Li et al. [14] investigated the genes of an aerobic denitrifying bacterium, *Acinetobacter* sp. YT03, that was associated with nitrite removal. They suggested that, among the DEGs, C4-dicarboxylate transporter (DctA) and nitrate/nitrite transporter (Nrt) might be important for utilizing carbon and nitrogen sources. To the best of our knowledge, there is a lack of in-depth, mechanistic understanding of how nitrifying bacteria regulate cellular networks in response to ammonia nitrogen stress.

In this study, the ammonia nitrogen removal ability of an aerobic nitrifying bacterium, *Klebsiella* sp. TN-10, was investigated under different nitrogen stress levels. The aim of this study was to reveal the DEGs by RNA-Seq and describe the related regulatory network in response to ammonia nitrogen stress. The analysis of the regulatory network of strain TN-10 under ammonia nitrogen stress laid the foundation for understanding the response of nitrifiers to ammonia nitrogen stress.

## 2. Materials and Methods

### 2.1. Culture Condition and Ammonia Nitrogen Stress of Bacterial Strain

The strain *Klebsiella* sp. TN-10, which has great nitrification ability, was isolated from tannery wastewater and stored in the China Center for Type Culture Collection (CCTCCM2017193). The basic medium (BM) used for exploring the growth and nitrification performance contained the following components (per liter): 7 g sodium pyruvate, 0.5 g (NH_4_)_2_SO_4_, 0.4 g FeSO_4_·7H_2_O, 0.5 g MgSO_4_, 1 g K_2_HPO_4_, and 0.8 g EDTA, at pH 7.0. To study the effects of ammonia nitrogen stress on the growth and nitrification ability of the isolate, 1 mL (1% inoculum) seed broth cultured for 12 h (150 rpm, 30 °C) was inoculated into the BM with different initial (NH_4_)_2_SO_4_ concentrations (0.5, 2.5, and 5 g/L). The cells were cultivated in a shake flask (150 rpm, 30 °C), and the samples were harvested by centrifugation (10,000 rpm, 5 min) at 6 h intervals. The supernatant was used to determine the concentration of ammonia nitrogen, and the pellet was used to determine the OD_600_. All of the above experiments were conducted in triplicate.

### 2.2. DNA Extraction and Genome Sequencing

After cultivation for 12 h (150 rpm, 30 °C), cells were collected to extract genomic DNA using a ChargeSwitch^®^ gDNA Mini Bacteria Kit (Invitrogen, Carlsbad, CA, USA) according to the manufacturer’s instructions. The purified genomic DNA was sequenced using a combination of PacBio RSII and Illumina sequencing platforms. Canu and SOAPdenovo software were used to assemble the PacBio and Illumina reads to produce the complete genome sequence. Then, a circular genome sequence without the existing gap was generated. Glimmer version 3.02 (http://cbcb.umd.edu/software/glimmer/) was used to identify the open reading frames (ORFs). The ORFs with more than 300 base pairs were queried against non-redundant (NR) frames in the NCBI database, SwissProt database (http://uniprot.org), KEGG database (http://www.genome.jp/kegg/), and COG database (http://www.ncbi.nlm.nih.gov/COG) to perform functional annotation.

### 2.3. RNA Extraction

After cultivation for 12 h (150× *g* rpm, 30 °C) in different (NH_4_)_2_SO_4_ concentrations, cells were collected, and the triplicate samples of the total RNA were extracted with TRIzol Reagent (Invitrogen, Waltham, MA, USA). A NanoDrop 2000 spectrophotometer (NanoDrop Technologies; Wilmington, DE, USA) was used to determine the concentration and purity of the RNA. The RNA integrity was measured using 1.5% agarose gels, and the RNA integrity number (RIN) was determined by a 2100 Bioanalyzer (Agilent Technologies, Santa Clara, CA, USA). High-quality RNA samples (OD260/280 = 1.8~2.0, RNA concentration of ≥100 ng/μL) were used to construct a sequencing library.

### 2.4. Library Construction and Illumina Deep Sequencing

Ribosomal RNA (rRNA) depletion was performed to isolate mRNA, which was then broken into short fragments (200 bp). The first-strand cDNA was synthesized using short fragments as templates. Then, the second-strand cDNA was generated and subjected to end-repair, phosphorylation, and adenine (A) nucleotide addition of the 3’ ends. The cDNA was ligated to adaptors, and the cDNA target template of 200 bp was amplified using Phusion DNA polymerase (NEB) for 15 PCR cycles. After quantification by a TSB380 Fluorometer (Invitrogen), the purified libraries were sequenced using an Illumina HiSeq X Ten System (2 × 150 bp read length) at Shanghai Majorbio Bio-pharm Biotechnology Co., Ltd. (Shanghai, China).

### 2.5. RNA-Sequencing, de Novo Transcriptome Assembly

The low-quality reads and adaptors were removed using SeqPrep software (https://github.com/jstjohn/), and the high-quality clean reads for de novo assembly were conducted using Trinity (https://github.com/trinityrnaseq/trinityrnaseq/wiki). Clean reads were broken into shorter reads (K-mers) and connected to longer reads (contigs) by integrating the overlaps. These contigs were clustered into genes by mate pair joining using Illumina. The assembled genes were annotated and analyzed.

### 2.6. Differentially Expressed Gene Analysis and Functional Enrichment

The clean reads were aligned with the genome of Klebsiella sp. TN-10 using Bowtie 2 (http://bowtie-bio.sourceforge.net/bowtie2/index.shtml). The fragments per kilobase per million fragments mapped (FPKM) method was used to estimate the gene expression levels [15]. The differently expressed genes (DEGs) among the control and ammonia stress treatment groups were detected using DESeq software with the q-value threshold restricted to q < 0.05 and log2 (fold change) > 1. Gene Ontology (GO) functional enrichment analysis was implemented to identify the GO terms and determine the biological function of the DEGs. Goatools (https://github.com/tanghaibao/GOatools) was used to identify the statistically significantly enriched GO terms using Fisher’s exact test. To explore the most important biological metabolic pathways and signal transduction pathways of the DEGs, the Kyoto Encyclopedia of Genes and Genomes (KEGG) pathway enrichment analysis was performed using KOBAS 2.0 (http://kobas.cbi.pku.edu.cn/home.do). The raw RNA-seq data were submitted to the NCBI under the accession number PRHNA588654 (https://www.ncbi.nlm.nih.gov/sra/PRJNA588654).

### 2.7. Validation of the RNA-Sequencing Data by Quantitative RT-PCR

For quantitative RT-PCR analysis, five differentially expressed genes were selected. RNA was isolated using an RNA extraction kit (Takara Bio, Inc., Japan). Complementary DNA was synthesized using a cDNA Synthesis Kit (Takara Bio, Inc., Japan). The primers used for the RT-PCR assay are listed in Appendix A. The 16S rRNA was used as the internal control for quantification. The RT-PCR assay was performed using a SYBR Premix Ex TaqTM Kit (Takara Bio, Inc., Japan) with at least three biological replicates. The PCR was conducted with a LightCycler 480 II Real-Time PCR System (Roche, Germany) using the following protocols: 95 °C for 5 min, followed by 40 cycles at 95 °C for 5 s, 60 °C for 20 s, and a final cooling step at 50 °C for 30 s. The 2^−^^ΔΔC^_T_ method was used to compare the expression of the genes, and the expression levels of all the tested genes were normalized against the expression level of the internal control gene (16S rRNA).

## 3. Results and Discussion

### 3.1. Effects of Initial Ammonia Nitrogen Concentration on the Growth and Ammonia Nitrogen Removal of Klebsiella sp. TN-10

The media with the initial (NH_4_)_2_SO_4_ concentrations of 0.5, 2.5, and 5 g/L were defined as group-L, group-M, and group-H, respectively, and the effects of different ammonia nitrogen levels on growth and ammonia nitrogen removal were investigated (Figure 1). As shown in Figure 1a, the changes in biomass under different (NH_4_)_2_SO_4_ stress levels were very similar from 0–48 h. Figure 1b shows that the total amount of ammonia nitrogen degradation increased with the increase in initial ammonia nitrogen concentration. At the mid-log phase (12 h), the removed ammonia nitrogen amounts under different (NH_4_)_2_SO_4_ stress levels (group-L, group-M, and group-H) were 70.28, 118.33, and 157.18 mg/L, respectively. After 24 h, the ammonia nitrogen in group-L was completely removed, and the total removal of ammonia nitrogen in group-M and group-H did not change significantly. This demonstrated that the cells under greater ammonia nitrogen stress removed more ammonia nitrogen. In order to explore the effects of (NH_4_)_2_SO_4_ stress on ammonia nitrogen removal at the level of mRNA, the cells were collected at the mid-log phase (12 h) for transcriptome sequencing.

### 3.2. General Genome Information of Strain TN-10

The genome of *Klebsiella* sp. TN-10 has a total of 6,036,348 bp with an average GC content of 55.67% (Appendix A). The total length of the CDs (coding sequence) is 5,289,228 bp with an average length of 907 bp. The genome contains a circular chromosome (5,748,095 bp) and two plasmids with sizes of 193,950 bp and 94,303 bp. Eighty-four tRNA and forty-five rRNA were identified in the genome. The basic characteristics of the genome are shown by the genomic circle map (Appendix A).

### 3.3. De-Novo Assembly and DEGs Analysis

A total of 21,627,024, 20,642,886, and 21,669,372 raw reads were obtained from group-L, group-M, and group-H, respectively (Appendix A). After clean-up and quality filter, 21,104,885, 20,162,418, and 21,108,693 clean reads were obtained with a raw error rate of less than 1% for each sample. The percentages of Q20 and Q30 bases were higher than 98% and 95%, respectively. These results indicate that the data are high-quality and can be used for further analysis.

All transcripts in group-M and group-H were compared with those in group-L with a *p*-value ≤ 0.05 and log2 (fold change) ≥ 1. A comparison between group-M and group-L showed that a total of 23 significant DEGs were identified with 9 up-regulated genes and 14 down-regulated genes (Figure 2a,b). A comparison between group-H and group-L identified 1209 DEGs with 538 genes up-regulated and 671 genes down-regulated (Figure 2c,d). These results showed that the number of DEGs in group-L vs. group-M was significantly lower than the number in group-L vs. group-H. Moreover, the changed magnitude in the levels of DEGs in group-L vs. group-H was significantly higher than the levels in group-L vs. group-M. These gene expression patterns demonstrated that higher ammonia nitrogen concentration had a greater influence on the gene expression of the TN-10strain.

### 3.4. Enrichment Analysis of DEGs

Similar to the above selection conditions (*p <* 0.05 and log2FC ≥ 1), these DEGs were classified into three gene ontology (GO) categories: biological process (BP), cellular component (CC), and molecular function (MF). As shown in Figure 3a, the number of down-regulated genes was higher than the number of up-regulated genes. Throughout the various subcategories of BP, the highest numbers of DEGs were observed in metabolic process and cellular process. There were a few genes within the CC subcategories. In the MF category, 5 up-regulated genes and 10 down-regulated genes were enriched in the catalytic activity subcategory, and 3 up-regulated genes and 4 down-regulated genes were observed in the binding subcategory.

Figure 3b shows that the number of DEGs in group-H was much higher than the number in group-M. In BP, the largest subcategories of DEGs were metabolic process and cellular process. In the metabolic process subcategory, 219 up-regulated DEGs and 257 down-regulated DEGs were identified, while 209 up-regulated DEGs and 217 down-regulated DEGs were identified in the cellular process subcategory. Within the CC category, the two most frequent subcategories were cell part and membrane part, with more than 100 DEGs. In the MF category, 261 up-regulated DEGs and 361 down-regulated DEGs were enriched in the catalytic activity category, while 210 up-regulated DEGs and 243 down-regulated DEGs were identified in the binding category. In addition, there were a few DEGs enriched in the transporter and transcription regulator activity subcategories. These results showed that more DEGs were identified under high ammonia nitrogen stress, and suggests that these DEGs may participate in more pathways of secondary metabolites and regulate a complex mechanism in ammonia nitrogen degradation.

KEGG is a database used to analyze the enriched metabolic pathways of DEGs [16]. As shown in Figure 4a, 16 DEGs in group-M were assigned to 13 known pathways, which were divided into three categories: metabolism, environmental information processing, and cellular processes. Within the metabolism category, six DEGs (gene2714, *sad*; gene0845, *gcvp*; gene4851, *hpaD*; gene4852, *hpaF*; gene3316, *astD*; and gene3314, *astE*) participated in amino acid metabolism. In the cellular processes category, the most abundant subcategory, cellular community-prokaryotes, was involved in six DEGs (gene1826; gene0651, *IsrK*; gene0656, *IsrB*; gene1827; gene1824; and gene1825).

As shown in Figure 4b, DEGs in group-H participated in 141 pathways. These pathways were divided into six categories: metabolism, genetic information processing, environmental information processing, cellular processes, organismal systems, and human diseases. In the metabolism category, the most abundant subcategory was carbohydrate metabolism with 167 DEGs, followed by amino acid metabolism (112 DEGs), energy metabolism (82 DEGs), and metabolism of cofactors and vitamins (55 DEGs). Within the environmental information processing category, 110 DEGs were involved in membrane transport and 53 genes were involved in signal transduction. As predicted, the DEGs under higher ammonia nitrogen stress participated in more pathways. Of these pathways, carbohydrate metabolism and amino acid metabolism were the most abundant subcategories, which could be related to the higher ammonia nitrogen degradation in group-H.

### 3.5. Function Enrichment of DEGs

Figure 5 shows the significantly enriched KEGG pathways of DEGs. Due to the involvement of DEGs in group-H in many pathways, only five of the most-enriched function pathways are presented in the figure. As shown in Figure 5a, the five enriched pathways of DEGs under high ammonia nitrogen stress were: inositol phosphate metabolism; phosphotransferase system; oxidative phosphorylation; fatty acid degradation; and valine, leucine, and isoleucine degradation. In group-M, the main enriched pathways were quorum sensing, tyrosine metabolism, arginine and proline metabolism, glyoxylate and dicarboxylate metabolism, and bacterial chemotaxis. The left portion of the circle diagram corresponding to the function pathways represents the most-enriched pathways and the changed magnitude in the levels of the DEGs.

### 3.6. Effect of Ammonia Nitrogen Stress on the Main Pathway of Energy Supply

Figure 6 shows the carbon metabolism pathways in which the DEGs were involved. Due to few DEGs in group-M being identified and the multiple being low, only the comparison of DEGs between group-H and group-L is presented in the pathways. As shown in Figure 6, the expression of the gene2575 (*malX*, encodes phosphotransferase system maltose transporter) was log_2_ fold change (LFC) 2.19 times down-regulated, and the gene3830 (*bglA*, encodes 6-phospho-beta-glucosidase) was down-regulated by LFC 1.11. Glucose can be converted to pyruvate via the glycolysis pathway, in which 6-phosphogluconate is a key intermediate [17]. Pyruvate, as the end-product of glycolysis, plays an important role in ATP generation. Thus, the depression of the glycolysis pathway may influence the following tricarboxylic acids cycle (TCA) pathway.

In the whole TCA cycle, all DEGs were down-regulated. The gene3957 (*sucA*, encodes 2-oxoglutarate dehydrogenase) was LFC 2.61 times down-regulated, which plays a key regulatory role in the TCA cycle. In addition, the genes encoding dihydrolipoamide succinyltransferase (gene3956, *sucB*), succinyl-CoA ligase (gene3955, *sucC*), succinate dehydrogenase (gene3961, *sdhC*; gene3958, *sdhB*; and gene3959, *sdhA*), fumarate hydratase (gene1580, *fumC*), and malate dehydrogenase (gene0517, *mdh*) in the TCA cycle were significantly down-regulated. The down-regulation of genes in the TCA cycle showed that the energy supply was depressed for cells under high ammonia stress, although different results have been reported. For example, when *Haloarcula hispanica* was cultured in a nutrient-limited condition, the proteomic and transcriptomic analyses showed that the TCA cycle was down-regulated, possibly because excess carbon source was supplied and more energy was provided to maintain balanced growth [18]. In addition, it was reported that genes of *Nitrosomonas europaea* 19,718 involved in the TCA cycle were down-regulated under ammonia starvation [19].

Regarding the valine, leucine, and isoleucine degradation pathway, all DEGs were down-regulated, including gene1531 (*fadA*, encodes 3-ketoacyl-CoA thiolase), gene4513 (*fadE*, encodes acyl-CoA dehydrogenase), gene2219 (*fadD*, encodes long-chain-fatty-acid-CoA ligase), gene1532 (*fadJ*), and gene5356 (*fadB*, both *fadJ* and *fadB* encode multifunctional fatty acid oxidation complex subunit alpha). Valine, leucine, and isoleucine are branched-chain amino acids (BCAAs), and play important roles in the organism [20]. In addition, it has been reported that BCAA participated in lipolysis, lipogenesis, glucose metabolism, glucose transportation, etc. [21,22]. Meanwhile, the acetyl-CoA and succinyl-CoA resulting from leucine, valine, and isoleucine entered the TCA cycle, which may have been related to the depression in the TCA cycle. Finally, similar to the above pathways, the pathway of fatty acid degradation was down-regulated under high ammonia nitrogen stress, which may be ascribed to the depressed TCA cycle.

The LEfSe analysis of the DEGs shows that both gene1532 (*fadJ*) and gene5063 (*mmsA*) were more important in the fatty acid degradation within group-L. Additionally, gene4513 (*fadE*) and gene5356 (*fadB*) were more important in the valine, leucine, and isoleucine degradation within group-M.

Figure 7 shows the effects of ammonia nitrogen stress on the expression of genes involved in oxidative phosphorylation. As is shown in Figure 7, almost all the DEGs were down-regulated. Genes related to encoding NADH dehydrogenase 1588–1597 (*nuo*) were down-regulated by LFC 1.02–1.26. Genes 3958–3961 (*sdh*, encode succinate dehydrogenase) were down-regulated by LFC 2.23–2.95. Additionally, genes 4311–4315 (*cyt*, encode cytochrome c oxidase) were down-regulated by LFC 1.65–2.67. Oxidative phosphorylation is a primary way to provide ATP through the respiratory chain [23]. The results showed that ammonia nitrogen in high concentrations suppressed the process of oxidative phosphorylation in cells. Finally, when combining the process of glycolysis, the TCA cycle, and oxidative phosphorylation, the activities supporting energy were suppressed, which may be ascribed to the toxicity of high-concentration ammonia nitrogen on cells.

### 3.7. Effect of Ammonia Nitrogen Stress on Amino Acid Metabolism

As shown in Figure 8, we analyzed the DEGs of the ornithine cycle, which is the crucial pathway in amino acid metabolism. The amount of up-regulated DEGs showed that the urea cycle was accelerated under ammonia nitrogen stress (group-H). Gene 5033 (*argF*, encodes ornithine carbamoyltransferase) up-regulated by LFC 3.52, which quickly converts carbamoyl-P and ornithine to citrulline. Gene0576 (*agrG*, codes for argininosuccinate synthase) up-regulated by LFC 1.41. Argininosuccinate synthase can not only convert citrulline to l-argininosuccinate lyase but also breaks down aspartate to a variety of intermediate metabolites that participate in the glycine, serine, and threonine metabolism. Gene5463 (*argH*, encodes argininosuccinate lyase) up-regulated by LFC 1.76 and accelerated the conversion of l-arginosuccinate to arginine in the urea cycle. Gene3317 (*astA*, codes for arginine N-succinyltransferase) was down-regulated by LFC 4.65, which inhibits the conversion from arginine to N2-succinyl-L-arginine. In the whole metabolism process of arginine and proline, all DEGs are down-regulated, which causes the whole metabolism pathway to weaken. Therefore, arginine may have accumulated, and the whole urea cycle accelerated. Generally, excreting ammonia nitrogen in the form of urea is an important mode of ammonia detoxification [24]. It was reported that *Litopenaeus vannamei* adapted to high-concentration ammonia stress through excreting ammonia [25]. Some bacteria fought against acid stress by using urease and arginine deiminase for ammonia accumulation [26].

In this study, the cells responded to ammonia nitrogen stress by converting ammonia to urea and excreting it to the extracellular matrix. In fact, arginine, proline, and glutamine are able to transform mutually, which plays an important role in ammonia nitrogen metabolism regulation, immune response, and biological growth [27]. In addition, proline is able to stabilize the intracellular environment, protect protein integrity, increase nitrate reductase activity, reduce lipid peroxidation, and protect the plasma membrane [28]. Phenylalanine, tyrosine, and tryptophan can catalyze to acetyl-CoA, which participates in the TCA cycle and releases energy [29]. In this study, the accelerated urea cycle and depressed arginine and proline metabolism under ammonia nitrogen stress may be an important factor in the increase in total ammonia nitrogen degradation. The up-regulated DEGs in the amino acid metabolism can cause cells to produce more metabolites for other life activities. In addition, the metabolic process of amino acid can provide more intermediates for cell metabolism. We suggest that the acceleration of amino acid metabolism plays an important role in promoting ammonia nitrogen degradation under ammonia nitrogen stress.

### 3.8. Effect of Ammonia Nitrogen Stress on the Nitrogen Cycle Pathway

The nitrogen metabolism of cells under ammonia nitrogen stress was analyzed, and the DEGs related to the nitrogen cycle are shown in Figure 9. Since the genes in group-M did not express differentially, only the DEGs in group-H are presented in the nitrogen cycle. As shown in Figure 9, gene0533 (*gltB*, encodes glutamate synthase) was up-regulated by LFC 2.25. It was reported that *gltB* participates in ammonia assimilation and regulation [30]. Differing from our results, the *gltB* in *Haloarcula hispanica* was induced in a nitrogen-limited medium [18]. Additionally, the genes involved in nitrification pathways were not differentially expressed, and some DEGs were up-regulated in the denitrification pathways. Gene2312 (*narG*, codes for nitrate reductase subunit alpha), gene2313 (*narH*, codes for nitrate reductase subunit beta), and gene2315 (*narI*, codes for respiratory nitrate reductase subunit gamma) were up-regulated by LFC 2.95, 3.26, and 2.69, respectively. The nitrate reductase encoded by *narG* and *narH* is the key enzyme in nitrate-respiring [31], and the accelerated denitrification pathways in the TN-10 strain led to a higher degradation of ammonia nitrogen. Meanwhile, gene2311 (*narK*, codes major facilitator superfamily transporter and relates to nitrate/nitrite transmembrane transporter activity) was up-regulated by LFC 2.57 [32]. The up-regulation of *narK* in group-H accelerated the entry of ammonia nitrogen into the intracellular region and expedited the utilization of ammonia nitrogen indirectly. The increase in ammonia nitrogen degradation in group-H may be ascribed to increased transmembrane transporters and accelerated denitrification pathways.

### 3.9. q-PCR

q-PCR is a precise, sensitive, and simple procedure, which has become the method for the detection and quantification of mRNA. To further validate the data obtained by RNA-sequencing, q-PCR analysis was performed by selecting five differentially expressed genes involved in nitrogen metabolism and oxidative phosphorylation (Appendix A). Interestingly, the q-PCR results of all selected genes exhibited a positive correlation with the RNA-sequencing data.

## 4. Conclusions

The ammonia nitrogen removal ability of the heterotrophic nitrification bacterium *Klebsiella* sp. TN-10 was determined, and the results showed that the strain removed more ammonia under higher ammonia nitrogen stress. RNA-Seq transcriptome analysis showed that some physiological aspects of the TN-10 strain are modified under ammonia nitrogen stress. Key metabolic pathways that DEGs are involved in were analyzed, including carbohydrate metabolism and amino acid metabolism. Several important DEGs in the nitrogen cycle were investigated. This study may help clarify the gene regulation of nitrifying bacteria under nitrogen stress.

## Figures and Tables

**Figure 1 microorganisms-10-00353-f001:**
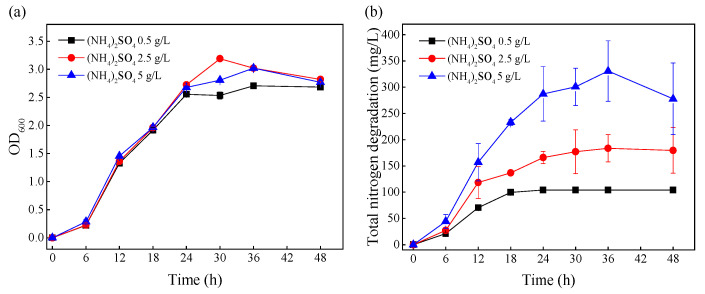
The effects of initial nitrogen concentrations on the growth and nitrogen removal of TN-10. Cells were cultivated in mediums with initial (NH_4_)_2_SO_4_ concentrations of 0.5, 2.5, and 5 g/L. (**a**) The change in OD600 over time; (**b**) the degradation of nitrogen over time.

**Figure 2 microorganisms-10-00353-f002:**
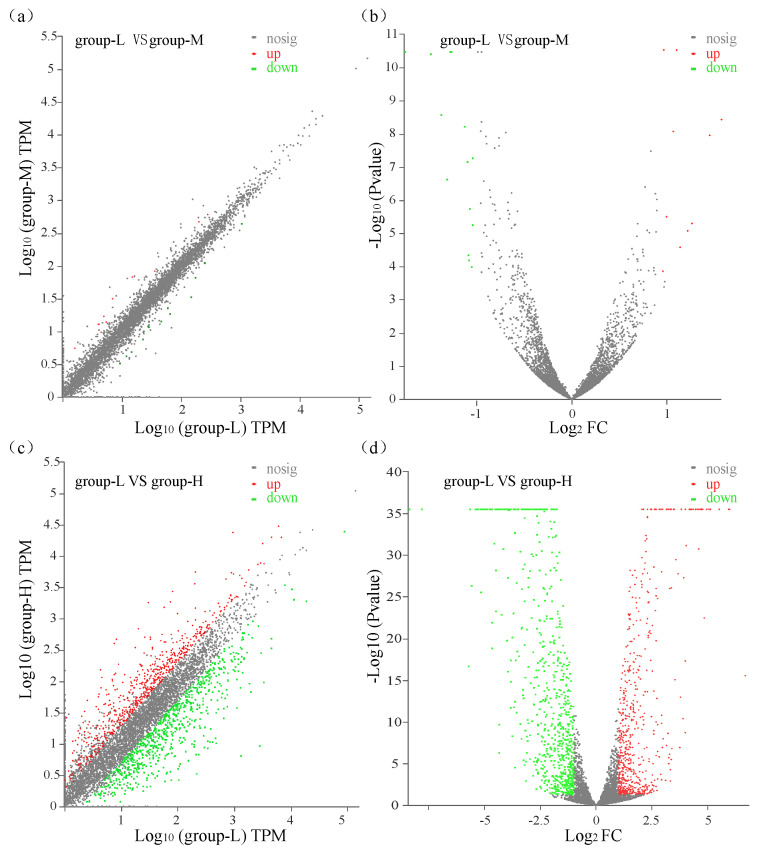
Scatter plots and volcano plots of DEGs that were identified. Green dots: down-regulated DEGs; red dots: up-regulated DEGs; gray dots: unchanged genes. (**a**) Scatter plot of DEGs compared between group-L and group-M; (**b**) volcano plot of DEGs compared between group-L and group-M; (**c**) scatter plot of DEGs compared between group-L and group-H; (**d**) volcano plot of DEGs compared between group-L and group-H.

**Figure 3 microorganisms-10-00353-f003:**
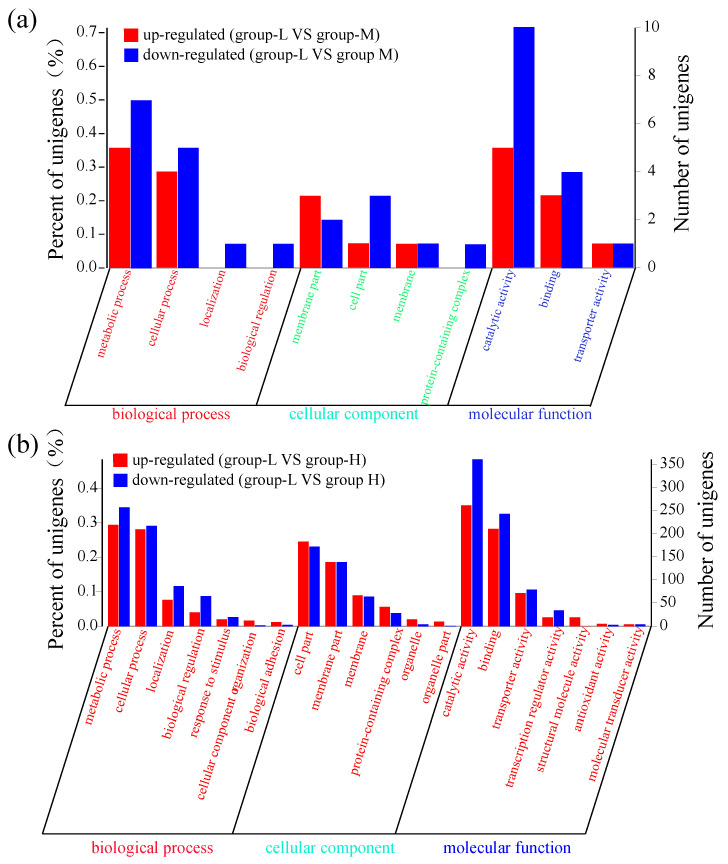
GO functional enrichment of DEGs. The *X*-axis represents the function of various genes. The *Y*-axis on the left represents the percent of genes. The *Y*-axis on the right corresponds to the number of DEGs. (**a**) DEG comparison between group-L and group-M; (**b**) DEG comparison between group-L and group-H.

**Figure 4 microorganisms-10-00353-f004:**
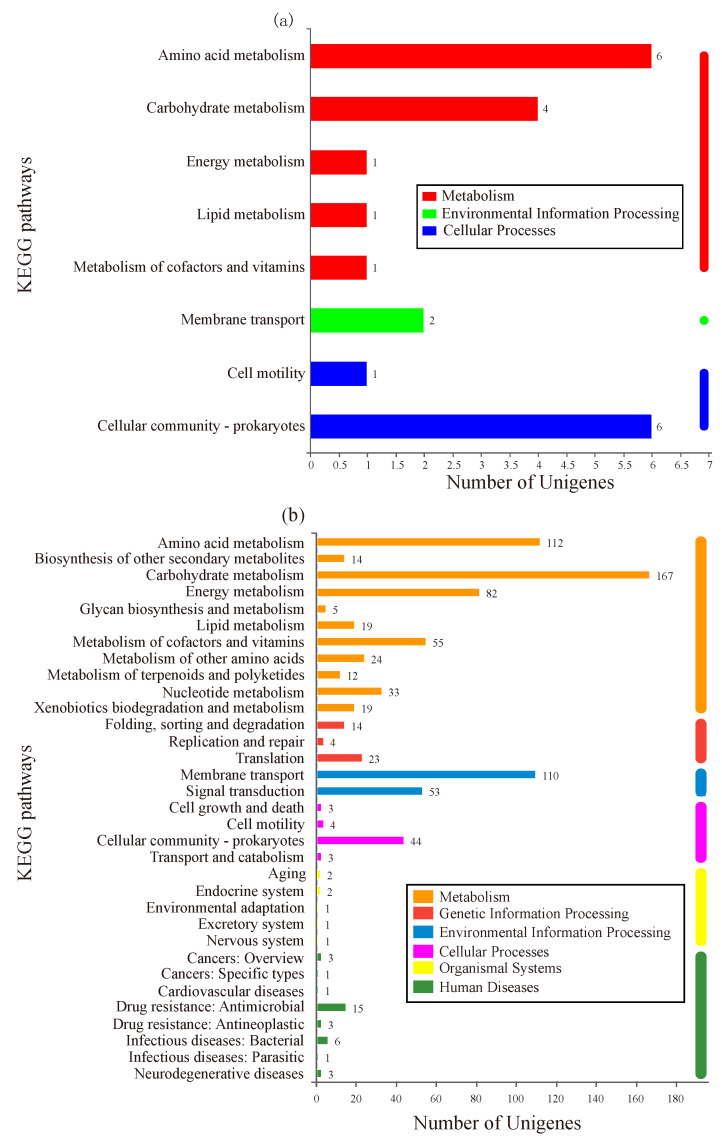
KEGG enrichment analysis of DEGs. The *X*-axis represents the number of genes. The *Y*-axis corresponds to the name of the pathway. (**a**) The cells were stressed under moderate nitrogen (group-M). (**b**) The cells were stressed under a high concentration of nitrogen (group-H).

**Figure 5 microorganisms-10-00353-f005:**
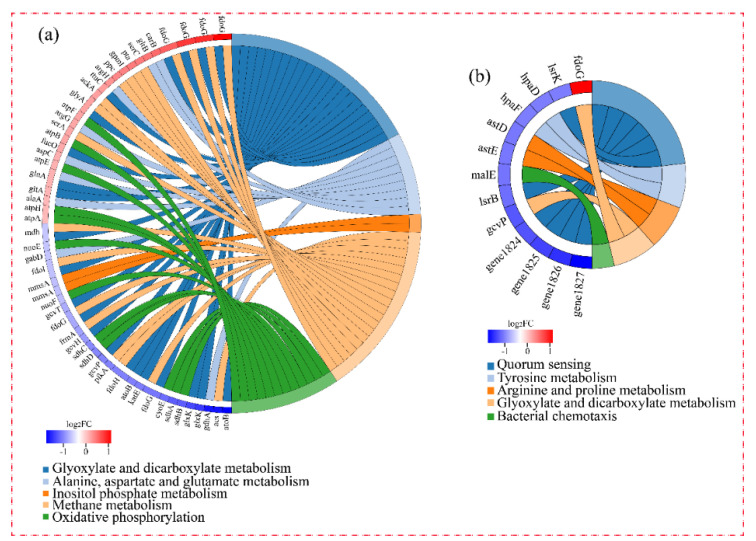
KEGG functional enrichment string diagram. The left portion of the circle diagram represents the DEGs, which are arranged by log_2_FC in descending order. The larger the log_2_FC, the larger the magnitude of the up-regulated DEGs. The smaller the log_2_FC, the larger the magnitude of the down-regulated DEGs. The closer log_2_FC is to zero, the smaller the changed magnitude in the levels of the DEGs. The right portion of the circle diagram represents the significantly enriched KEGG pathways. The 5 most significant enrichment pathways are presented. (**a**) The DEGs of cells under high nitrogen stress. (**b**) The DEGs of cells under moderate nitrogen stress.

**Figure 6 microorganisms-10-00353-f006:**
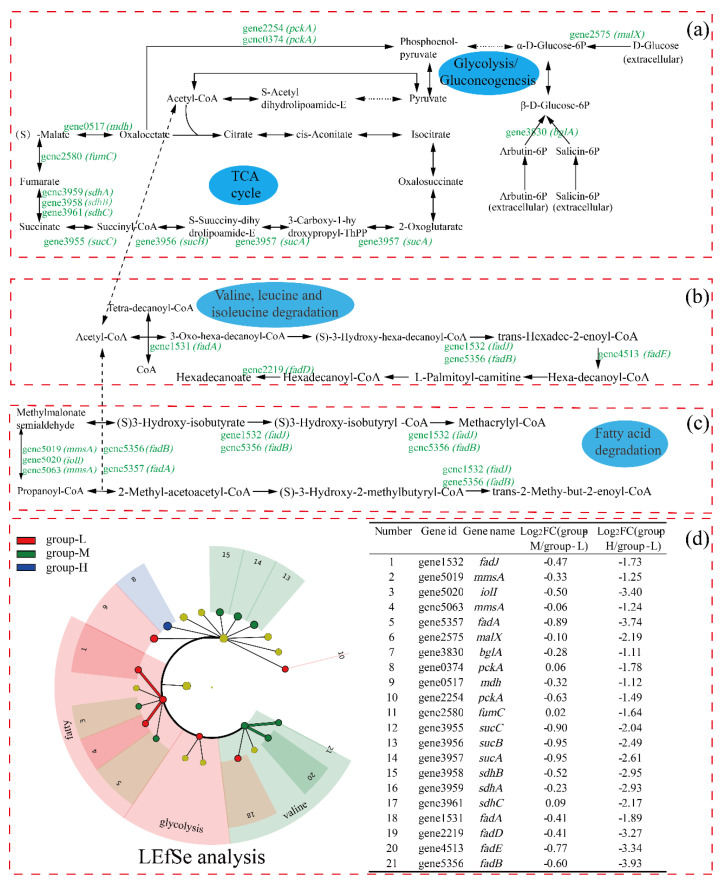
The effect of nitrogen stress on the expressions of genes involved in the TCA cycle: glycolysis (**a**); valine, leucine, and isoleucine degradation (**b**); and fatty acid degradation (**c**). Genes with up-regulated and down-regulated DEGs are presented in red and green, respectively. LEfSe was used to analyze the DEGs involved in the above pathways (**d**).

**Figure 7 microorganisms-10-00353-f007:**
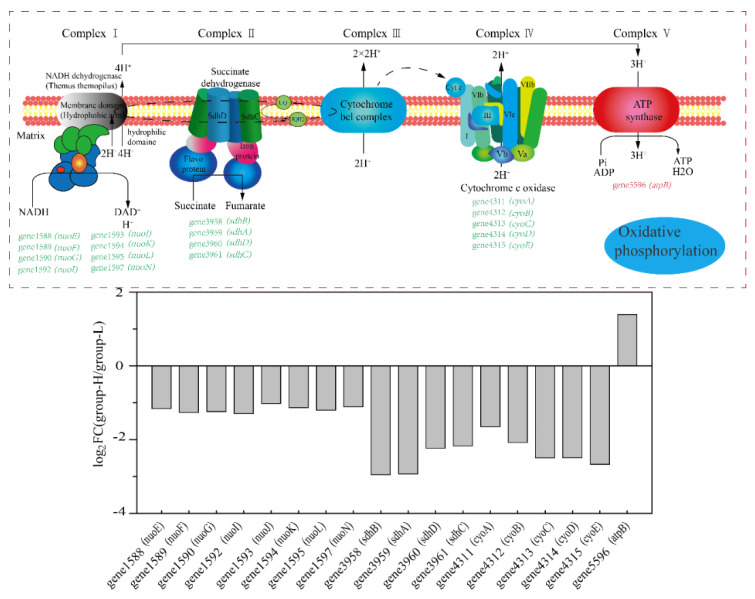
The effect of nitrogen stress on oxidative phosphorylation. The up-regulated and down-regulated genes in group-H compared to the genes in group-L are shown in red and green, respectively. The multiples in changes of the genes in the pathways are shown in the column diagram.

**Figure 8 microorganisms-10-00353-f008:**
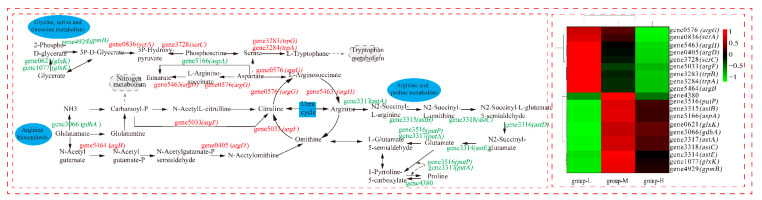
The effect of nitrogen stress on amino acid metabolism. The heatmap on the right shows the expression of the DEGs in the three groups, individually.

**Figure 9 microorganisms-10-00353-f009:**
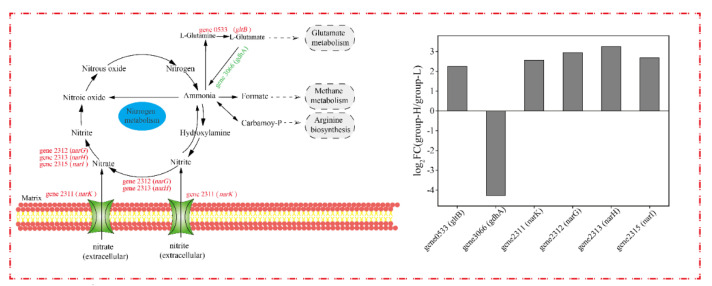
The effect of nitrogen stress on DEGs involved in the nitrogen cycle. The up-regulated and down-regulated genes in group-H compared to group-L are shown in red and green, respectively.

## Data Availability

The raw RNA-seq data generated in this study were submitted to the NCBI.

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
