# Peer review of "Comprehensive Transcriptomic Analysis of Heterotrophic Nitrifying Bacterium *Klebsiella* sp. TN-10 in Response to Nitrogen Stress"

_microorganisms, 2022, doi:10.3390/microorganisms10020353_

Round 1

Reviewer 1 Report

Authors herein provide a fine description of the effects of nitrogen starvation at transcriptome level in a strain of K. In my opinion the manuscript is well organised including the experimental and results  section. also, the figure are of good quality and report sufficient information for the reader. It is notheworty that the authors carried out also genome sequencing via PacBio. Only an issue should be addressed. Why the authors do not validate Illumina seq via qPCR on a few DEG?

Only an issue should be addressed

Reviewer 2 Report

The paper reports interesting and new results of the changes in gene expressions responding to high ammonium concentrations in a heterotrophic bacterial strain isolated from highly eutrophic environments. Significant changes in the gene expressions were observed in carbon metabolism, energy metabolism, amino acid metabolism, and denitrification metabolism. These findings are important for understanding the adaptation of bacteria to eutrophic and ammonium-rich environments. In addition, the information is potentially useful for improving wastewater treatment processes. I recommend publication the manuscript in Microorganisms after proper amendments following the suggestions below.

The word usage of “nitrifying bacteria” and “nitrogen stress” confused the reviewer in the first skim reading and possibly for many potential readers. The “nitrifying bacteria” is usually used for bacteria that oxidize ammonium to nitrites and further to nitrates. In the case of the bacterial strain used, the results showed denitrification co-occurs, and the accumulation of nitrates or nitrates was not shown. In that case, the process of nitrification should proceed in the bacteria, but people usually do not call them “nitrifying bacteria.” The second word, “stress,” is usually used for the situation of something difficult or uncomfortable for the organisms in biology. But the “nitrogen stress” of 5 g/L of ammonium sulfate in this paper was not shown to cause any difficult or uncomfortable situation for the bacteria. The growth was not inhibited or affected under the conditions, and the gene expression changes were mostly in the range of usual metabolic response but did not correspond to so-called “stress response.” I highly recommend avoiding the usage of “nitrifying bacteria” and “nitrogen stress” in the title and many parts of the text.

The following sentence was strange since “essential amino acids” for bacteria cannot be defined in general but should be shown individually for each strain. In addition, all amino acids play important roles in organisms.

L.285-287. Valine, leucine, and isoleucine are branched-chain amino acids (BCAA) which are essential amino acids and play important roles in the organism.

The following sentence was unclear for the reviewer since any obvious effect of the higher concentrations of ammonium sulfate is not shown in the growth of Figure 1.

L.306-307. The depression of oxidative phosphorylation would limit the excess growth of cells under ammonia nitrogen stress.

The followings are possible typing and grammatical errors that authors may consider for the revision.

L.17. which mean these: which means these

L.50-51. the depressed of: the depression of

L.52. bacterium Acintobacter sp.: bacterium Acinetobacter sp.

L.90. different NaCl concentrations: different (NH4)2SO4 concentrations

L.127. analysis were performed: analysis was performed

L.141-142. removal ammonia nitrogen: removal of ammonia nitrogen

L.159. reads obtained from: reads were obtained from

L.169. genes un-regulated and: genes up-regulated and

L.188. the 5 un-regulated genes: the 5 up-regulated genes

L.222. DEGs involved in: DEGs are involved in

L.223. genes involved in: genes are involved in

L.236. stress mainly involved: stress are mainly involved

L.253. group-M were identified: group-M being identified

L.253. multiple is low: multiple being low

L.269. ammonia stress. While: ammonia stress, while

L.298. Figure 7 showed the: Figure 7 shows the

L.301-302. succinate dchydrogenase were: succinate dehydrogenase were

L.319-320. carbamoy-P and omithine to citruline: carbamoyl-P and ornithine to citrulline

L.332. convert citruline to: convert citrulline to

L.185. help clarity the: help clarify the
